# The Hunger Games: Homeostatic State-Dependent Fluctuations in Disinhibition Measured with a Novel Gamified Test Battery

**DOI:** 10.3390/nu13062001

**Published:** 2021-06-10

**Authors:** Katharina Voigt, Emily Giddens, Romana Stark, Emma Frisch, Neda Moskovsky, Naomi Kakoschke, Julie C. Stout, Mark A. Bellgrove, Zane B. Andrews, Antonio Verdejo-Garcia

**Affiliations:** 1Turner Institute for Brain and Mental Health, School of Psychological Sciences, Monash University, Clayton, VIC 3800, Australia; katharina.voigt@monash.edu (K.V.); emily.giddens@monash.edu (E.G.); emmafrisch@gmail.com (E.F.); skovsky@gmail.com (N.M.); naomi.kakoschke@csrio.com.au (N.K.); julie.stout@monash.edu (J.C.S.); mark.bellgrove@monash.edu (M.A.B.); 2Department of Physiology and Biomedicine Discovery Institute, Monash University, Clayton, VIC 3800, Australia; romana.stark@monash.edu (R.S.); zane.andrews@monash.edu (Z.B.A.)

**Keywords:** cognitive impulsivity, hunger, homeostatic state, gamified cognitive battery

## Abstract

Food homeostatic states (hunger and satiety) influence the cognitive systems regulating impulsive responses, but the direction and specific mechanisms involved in this effect remain elusive. We examined how fasting, and satiety, affect cognitive mechanisms underpinning disinhibition using a novel framework and a gamified test-battery. Thirty-four participants completed the test-battery measuring three cognitive facets of disinhibition: attentional control, information gathering and monitoring of feedback, across two experimental sessions: one after overnight fasting and another after a standardised meal. Homeostatic state was assessed using subjective self-reports and biological markers (i.e., blood-derived liver-expressed antimicrobial protein 2 (LEAP-2), insulin and leptin). We found that participants who experienced greater subjective hunger during the satiety session were more impulsive in the information gathering task; results were not confounded by changes in mood or anxiety. Homeostatic state did not significantly influence disinhibition mechanisms linked to attentional control or feedback monitoring. However, we found a significant interaction between homeostatic state and LEAP-2 on attentional control, with higher LEAP-2 associated with faster reaction times in the fasted condition only. Our findings indicate lingering hunger after eating increases impulsive behaviour via reduced information gathering. These findings identify a novel mechanism that may underpin the tendency to overeat and/or engage in broader impulsive behaviours.

## 1. Introduction

Hunger and satiety states influence the cognitive systems that orchestrate the trade-off between impulsive and reflective responses [1]. Hunger increases impulsive behaviours across non-human animal species, constituting an adaptive mechanism that fosters fast responses to correct homeostatic imbalance [2,3,4]. In contrast, in humans, the nature and direction of hunger effects on impulsivity have not yet been established [5]. Cognitive impulsivity or disinhibition refers to the tendency to engage in rapid reactions to internal or external triggers without sufficient consideration of consequences [6,7]. When people experience hunger, this tendency could serve to facilitate fast, advantageous responses toward restoring homeostasis, similar to other animals [8]. However, it could also contribute to disadvantageous choices in daily life (e.g., failing to stop at a red light on your way to the food court) and to the maintenance of mental disorders involving episodes of fasting or malnourishment (e.g., eating disorders or addictions [5,9]). These positive and negative ramifications of hunger effects (i.e., leading to both positive and negative repercussions) make the relationship between homeostatic input and impulsivity more complex in humans than in non-human animals. This complexity has been captured in recent studies indicating that satiety only increases impulsive responding on specific cognitive tasks, such as those measuring reflection or motor impulsivity [10,11]. Therefore, satiety could promote impulsive behaviour by relieving “homeostatic pressure” prior to action [11].

Disinhibition is a multifaceted construct, with at least three component cognitive processes: attention, information sampling, and the ability to monitor positive and negative outcomes [12,13]. The lack of comprehensive measures that are sensitive and specific in measuring these separate cognitive processes, and are repeatable and valid [14,15], has also likely influenced the inconsistent results of studies comparing impulsivity across hunger and sated states [11]. We recently developed a novel framework and a comprehensive, gamified measure (the Cognitive Impulsivity Suite or CIS) to assess the cognitive underpinnings of disinhibition, namely, attentional control, information gathering and feedback monitoring [16]. The CIS has the advantage of measuring separate cognitive processes via different tasks with equal perceptual and motor demands, which increases the measure’s test-retest reliability and validity. In particular, CIS tasks show significant associations with both trait and behavioural measures of impulsivity [16]. In this study, we administered the CIS in healthy participants across hunger and satiety states together with assays of circulating levels of insulin, leptin, and liver-expressed antimicrobial peptide 2 (LEAP-2), which is an antagonist of ghrelin action that is elevated by calorie consumption signaling satiety [17]. We hypothesised that hunger would increase specific facets of disinhibition (i.e., information-gathering), although our assumptions about the specific cognitive/biological mechanism/s impacted remained exploratory.

## 2. Materials and Methods

### 2.1. Design

We used a within-subjects design wherein participants underwent two assessment sessions: (1) after overnight fasting (“hunger”); or (2) after consuming a standardised filling breakfast (2300 kj; “satiety”). The meal consisted of 1 × slice of banana bread (1460 kJ, 5.5 g protein, 14.8 g fat, 48 g carbohydrates, 26 g sugars), 150 mL orange juice (270 kJ, 3.3 g protein, <1 g fat, 37.5 g carbohydrates, 31.2 g sugars), and 1 × tub of flavoured yoghurt (715 kJ, 3.6 g protein, 9.5 g fat, 17.7 g carbohydrates, 17.2 g sugars). Sessions started at 9:00 am, and the session order was counterbalanced. During each session, participants performed the CIS, and provided blood samples for appetite hormone assays (see Procedures). The protocol aligns with best-practice recommendations for food-related research [18]. Ethics approval for this study was granted by the Monash University Human Research Ethics Committee (MUHREC).

### 2.2. Participants

Thirty-four participants (59% women) aged 18–40 years (M = 26.26 years, SD = 6.85 years) were recruited via community advertisement. Inclusion criteria were having normal or corrected to normal vision, fluency in English, no allergies or intolerances that would impact consumption of the food provided in the satiety manipulation, no history of head trauma (e.g., traumatic brain injury), neurological (e.g., epilepsy) or metabolic impairments (e.g., diabetes), and no current mental health conditions. Exclusion criteria included cannulation contraindications (i.e., HIV, Hepatitis A, B or C diagnosis, low or high blood pressure, blood-thinning medication). Appendix A displays participants’ characteristics.

### 2.3. Measures

#### 2.3.1. The Cognitive Impulsivity Suite (CIS)

The CIS measures three cognitive mechanisms of disinhibition (i.e., attentional control, information gathering, and feedback monitoring/shifting) within a unified, gamified suite of tasks [16]. The three CIS tasks (Figure 1) have optimal test-retest reliability and criterion validity [16]. The tasks have equalised perceptual characteristics and response modalities (i.e., two-choice response options), and thus performance differences can be ascribed to the specific mechanisms of interest.

*Caravan Spotter (Information Gathering)*. This task measures information accumulation using a perceptual decision-making task with an initially ambiguous but progressively clearer target. It comprises four blocks of 60 trials (240 total). The task objective is to safely guide a trade caravan through the desert by identifying nearby dangers and obstacles. During each trial, participants are presented with an image at 50% pixilation that linearly disambiguates into one of two obstacles (e.g., a wagon or a horse). Participants select which obstacle the pixelated image will become by pressing keys A (e.g., for wagon) or L (e.g., for horse) within a trial response window of 2000 ms. The pairs of target stimuli change after each block. In-game points are awarded for both speed and accuracy, fostering impulsive errors in participants with lower information gathering thresholds [19]. The performance indices of the task are reaction time, the proportion of correct responses and fast identification errors (i.e., the number of times an incorrect response was made in <1000 ms).

We further classified performance during this task using Drift Diffusion Modelling [20]. This framework describes information accumulation through four parameters: the drift rate v, the boundary separation a, the prior decision bias z and the non-decision time τ. The drift rate (v) corresponds to information accumulation, which can be interpreted as a general measure of sensitivity to the relevant configurations. The boundary separation (α) can be interpreted as a decision threshold reflecting the amount of information required to trigger a choice. The mean ‘non-decision’ time parameter (τ) refers to the time taken for all processes occurring prior to (i.e., sensory encoding) and after the decision-making process (i.e., motor-response) [20,21]. The starting point bias (z) describes whether there is an evidence accumulation advantage to the correct response. We applied the Hierarchical Drift–Diffusion Modelling toolbox to obtain the parameters from the data (HDDM; [22]) (Appendix A for modelling details).

*Bounty Hunter (Attentional Control)*. This task measures attentional control using a modified cued Go/No-Go task (GNG; [23]) comprising four blocks of 60 trials (240 total). During each trial, participants are presented with a go stimulus (i.e., bandit) they must shoot by pressing the keyboard space bar, or a no-go stimulus (i.e., sheriff) they must avoid shooting. Trials have a response window of 700 ms, with go-stimuli presented on 75% of trials. Each trial is preceded by a cue signal based on time of day to indicate upcoming stimuli (i.e., dawn indicates ‘go’ trials; dusk indicates ‘no-go’ trials). A key modification from the original cued GNG task involves stimulus onset asynchronies defined as the duration between cue and stimulus presentation [23]. Traditional cued GNG tasks have short stimulus onset asynchronies (e.g., 300 ms, 400 ms). In contrast, the current measure uses very short stimulus onset asynchronies (50 ms, 200 ms) to challenge attentional control, and long stimulus onset asynchronies (1500 ms, 3000 ms) to foster attentional disengagement, defined as withdrawal of focus from an initial target [24]. The outcome variables are reaction time (for total responses, correct responses, and commission errors), proportion of hits (detecting a go stimulus), proportion of times the no-go stimulus is hit (false alarms or commission errors), and proportion of missed go stimuli (misses). 

We further classified participants performance in the Bounty Hunter task within the Signal Detection Theory framework [25], which explains decision-making in terms of uncertainty. SDT discriminates between two independent parameters: sensitivity (*d’*) and response criterion (*c*). The sensitivity *d’* describes the ability of an individual to differentiate between a sheriff (go signal) and a bandit (no-go signal) (equivalent to the distinction between signal and noise in classic SDT). The response criterion *c* describes an individual’s decision strategy ranging from liberal (<0), to unbiased (=0), to conservative (>0). Both parameters were calculated based on an individual’s hits (i.e., shooting a bandit) and false alarm rates (i.e., shooting a sheriff) (Appendix A).

*Prospectors Gamble (Monitoring of Feedback)*. This task measures monitoring of feedback through a modified probabilistic reversal learning task [26], comprising four blocks of 40 trials (160 total). During each trial participants must select one of two stimuli (i.e., gold prospectors) presented on-screen by pressing keys A (left-side prospector) or L (right-side prospector). Participants are instructed to determine and continuously select the lucky (i.e., correct) prospector by learning which stimulus is more likely to return positive reinforcement (i.e., in-game points) than negative reinforcement (i.e., in-game penalties) when selected. The correct prospector provides an 80:20 ratio of positive to negative feedback in the first two blocks and 70:30 in the second two blocks. Contingency reversals occur after each block, or 40 trials. Participants are not instructed when contingency reversals occur and must determine them based on changing ratios of positive to negative feedback. Each trial has a response window of 1000 ms, and prospectors randomly switch locations to avoid motor perseveration.

#### 2.3.2. Homeostatic State Measures

*Visual analogue scales*. We used a standard 10-cm visual analogue scale (VAS) to assess the subjective feeling of hunger. The participants had to rate the degree of hunger they felt in the current moment on a scale ranging from 1 (very low) to 100 (very high). Using an identical scale, we also assessed mood (e.g., rate your current mood) and anxiety (e.g., rate the degree of physical anxiety that you feel right now) as control variables. Brief VAS questionnaires have been shown to satisfactorily measure internal state [27,28]. 

*Blood hormones*. Plasma levels of insulin, leptin and LEAP-2 were obtained via fasted blood drawings from intravenous cannula in order to assess changes in physiological markers of homeostatic state. The complete blood drawing protocol is outlined in the Appendix A.

### 2.4. Procedure

Participants first completed an online eligibility survey. Eligible participants completed two 1.5-hour laboratory testing sessions (fasting and satiety) scheduled 7–10 days apart (M = 8.6 days, SD = 1.4) at Monash University. Figure 2 summarises the procedure of both testing sessions. Sessions involved fasted or sated CIS administrations, with session order counterbalanced across participants (24 completed fasted first, 28 completed sated first).

On the evening before each session (i.e., from 7:00 pm the night before the experimental sessions), participants were asked to refrain from strenuous exercise, caffeine, nicotine and alcohol consumption [29,30]. They were instructed to consume a light dinner between 6:00 pm and 7:00 pm and then fast until their scheduled session. To avoid the possibility of food anticipation modifying blood hormone levels [31], participants were informed that they may consume a meal during both, one or no sessions to control for this effect. All sessions commenced at 9:00 am. Participants completed a guided CIS demonstration at the start of session one to minimise practice effects, which can occur between the first and second administration of cognitive tasks [32].

Participants in both conditions were first cannulated intravenously using a standard protocol (Appendix A). Next, during the sated session, participants consumed a standardised 2300 kj meal of banana bread, yoghurt and orange juice (carbohydrate: 76.4 g of which 53.8 g are sugar, protein: 10.5 g, fat: 24 g). In the fasted session, meal-consumption was replaced with the physical measurements (i.e., participants’ weight, height, waist and hip circumference. After the satiety or fasting protocols, participants completed the CIS. To minimise the influence of assessor on task performance, participants conducted the tasks on their own while task engagement was monitored from an adjacent room using the screen sharing software TeamViewer version 14 (2019), TeamViewer AG, Göppingen, Germany. During each session, blood samples were obtained directly following cannulation and again following completion of the full CIS approximately 60 minutes later. VAS (i.e., subjective reports of hunger, anxiety and mood) were completed following the CIS demo, cannulation, meal consumption/physiological measurement, and the complete CIS (Figure 2). Note that blood samples and VAS were measured at specific time points across the experimental session; however, for data analyses outlined here, only the blood and VAS data at the last time point were considered, as we expected the difference between hunger and satiety to be the largest towards the end of each session (approximately 60 minutes after meal consumption / physiological measurement). Participants were reimbursed $75 following study completion.

### 2.5. Statistical Analyses

#### 2.5.1. Raw Data Cleaning

Three out of the 34 participants only completed one session, resulting in missing data for fasted (*n* = 2) and sated (*n* = 1) conditions. Final sample sizes for fasted and sated conditions were 32 and 33, respectively. Quality checks on this final data set were performed to ensure validity of cognitive data using Eisenberg et al.’s [14] criteria (Appendix A). Due to failure to meet these quality criteria, the sample sizes of completed CIS tasks were as follows: Bounty Hunter at *N* = 31, Caravan Spotter at *N* = 28, and Prospectors Gamble at *N* = 33.

#### 2.5.2. Homeostatic Manipulation

To examine if the fasting manipulation had worked as intended, we conducted paired sample t-tests for participants’ subjective hunger and satiety (measured with VAS) as well as blood hormone levels. Significance levels were Bonferroni-adjusted based on the number of total tests. As BMI is associated with differences in experienced hunger [33] and blood hormone levels [17,34,35], we further examined whether there was an interaction between BMI and subjective hunger as well as BMI and hormone levels across homeostatic state.

#### 2.5.3. Impact of Homeostatic State on Impulsivity

The effects of homeostatic state on CIS performance were assessed via generalised mixed effects modelling using the lm4 package in R [36]. We set up two models for each outcome variable of the CIS as follows: CIS outcome variable = b_0_ + b_1_Condition + b_1_HungerReport + b_3_Condition × HungerReport + b_4_Age + b_5_Gender + b_6_BMI + ID + e;
CIS outcome variable = b_0_ + b_1_Condition + b_1_Insulin + b_2_LEAP-2 + b_3_Leptin + b_4_Condition × Insulin + b_5_Condition × LEAP-2 + b_6_Condition × Leptin + b_7_Age + b_8_Gender + b_9_BMI + ID.

Independent variables (i.e., Condition, subjective hunger reports from VAS, insulin, LEAP-2, Leptin) were regressed on each outcome variable (e.g., reaction time or boundary parameter) using these model equations. All independent variables were mean-centered, and Condition was coded as 0 = Fasted and 1 = Sated. Age, Gender (coded as 0 = female, 1 = male) and BMI were included as covariates of no interest. The link function for each generalised mixed effect model was either a linear (for continuous CIS outcome variables; e.g. reaction times) or a logit function (for categorical CIS outcome variables, e.g., proportion of correct responses). Stepwise backwards model selection was applied and the model with the smallest Bayesian Information Criterion (BIC) was chosen. 

## 3. Results

### 3.1. Homeostatic Manipulation

Participants reported significantly higher VAS ratings of subjective hunger during the fasted (M = 68.44, SD = 19.02) versus the sated condition (M = 47.47, SD = 15.03), (*t* (31) = 6.12, *p* < 0.001), showing that the experimental manipulation was successful. Insulin levels were also significantly lower during the fasted condition (M = 2.55, SD = 2.99) compared to the sated condition (M = 11.65, SD = 11.51), (*t* (31) = −4.09, *p* < 0.001). No other blood hormones changed as a function of homeostatic state (Figure 3, Appendix A).

Linear mixed effects modelling revealed that the observed differences in subjective hunger and insulin levels across homeostatic state were not associated with subjective reports in mood and anxiety, nor the individual’s age, gender or BMI. The relationships between BMI, subjective hunger and blood hormones across homeostatic states are detailed in the Appendix A. Blood hormone levels at the beginning and at the end of the fasted and sated condition are reported in the Appendix A.

### 3.2. Impact of Homeostatic State on Disinhibition

#### 3.2.1. Information Gathering

During the sated relative to the fasting condition, participants with higher subjective hunger levels displayed a lower decision boundary during the information gathering task. Specifically, there was a significant interaction effect between homeostatic state and subjective hunger levels on a decreasing boundary parameter (*a*) of the winning DDM (Condition × Hunger effect: β = −0.43, SE = 0.18, *p* = 0.02, *R*^2^ = 0.14, Figure 4). There were no further relationships between homeostatic state and information accumulation processes. 

#### 3.2.2. Attentional Control 

There were no significant effects of the homeostatic manipulation on the key measure of impulsivity (i.e., commission errors). During the sated relative to the fasting condition, participants were more likely to respond even in the presence of no-go stimuli (Condition effect of criterion: β = −3.51, SE = 0.10, *p* < 0.001, *R*^2^ = 0.19). Furthermore, during the sated relative to the fasting condition, participants had higher sensitivity in detecting the target (i.e., shooting a bandit) from the non-target (i.e., shooting a sheriff; Condition effect of d’: β = 3.48, SE = 0.12, *p* < 0.001, *R*^2^ = 0.17). Both effects were independent of subjective hunger or any blood hormone levels. 

#### 3.2.3. Monitoring of Feedback

There were no significant effects of the homeostatic manipulation on the Prospector’s Gamble task. There was a trend for the interaction effect between condition and leptin levels: during the sated relative to the fasted condition, participants had had slightly higher leptin levels (Condition × Leptin effect: β = 0.03, SE = 0.01, *p* = .06, *R*^2^ = 0.02).

### 3.3. Impact of Appetite Hormones

Participants’ LEAP-2 levels across homeostatic state was significantly associated with reaction times in the Attentional Control task (Bounty Hunter; Appendix A). Specifically, the best fitting LME model, which included condition, LEAP-2 and their interaction as independent variables, revealed that lower LEAP-2 levels during the sated condition were associated with slower reaction times (Condition × LEAP-2 effect: β = −0.03, SE = 0.01, *p* = 0.003, *R*^2^ = 0.16). This effect persisted for reaction times during commission errors (Condition × LEAP-2 effect: β = −0.03, SE = 0.01, *p* = 0.004, *R*^2^ = 0.16) and correct responses (Condition × LEAP-2 effect: β = −0.04, SE = 0.01, *p* = 0.002, *R*^2^ = 0.18). Figure 5 illustrates these results.

## 4. Discussion

We found that subjective hunger levels during the satiety session were associated with reduced information gathering; those participants with lingering hunger levels in the satiety session acted more impulsively. Although homeostatic state did not influence impulsive responses linked to attentional control and monitoring of feedback, satiety relative to fasting was associated with better stimulus discriminability and more liberal response styles in the attentional task. Furthermore, reduced LEAP-2 levels (which facilitate ghrelin actions via reduced antagonism of the receptor) correlated with slower responses during the attentional control task.

Greater hunger during the satiety session led to less cautious decision-making and more errors on the information gathering task. Previous research had suggested that satiety could decrease information gathering, but the potential mechanism remained unclear [11,37]. Our results suggest that lingering hunger sensations after eating promote this impulsive tendency. This interaction may be explained by an “active inference” account. Following meal consumption, participants expect satiety sensations [38]. The discrepancy between the satiety expectation and perceived hunger may have encouraged impulsivity in an attempt to rid the aversive state of persistent hunger. This is consistent with addiction models of active inference; wherein impulsive drug consumption serves to alleviate withdrawal [39]. Alternatively, our finding could be explained by the optimal foraging theory, which suggests that an organism’s allocation of cognitive resources is dependent on available energy stores, with lower energy associated with greater risk-taking in attempt to move energy levels towards homeostasis [40]. This aligns with the observation of a higher decision-boundary in participants with lower hunger levels across both sessions. However, the combination of remaining hunger and greater available energy during the sated condition may reflect the relative risk of impulsive decision-making following food consumption. Higher energy allows for greater risk, as potential losses do not compromise energy balance [41].

Satiety relative to fasting was associated with better vigilance and a more liberal (but similarly accurate) response criterion in the attentional control task. We did not, however, find significant effects of satiety on measures of disinhibition, such as commission errors. These results are in alignment with previous findings indicating no effects of hunger/satiety on commission errors when using non-food targets [42,43]. Satiety effects on increased discriminability are consistent with past research, wherein satiety tends to improve attentional vigilance in speeded reaction time tasks [5,44]. Altogether, findings for attentional control suggest that satiety may increase confidence in rapid responses while reducing errors attributable to attentional lapses [45].

Compared to hunger, satiety was associated with no significant differences in the monitoring of feedback task. Monitoring of feedback consists of two components: set-shifting (the ability to shift attention between mental sets or tasks [46]), and sensitivity to reward/punishment (the degree to which behaviour is shaped by reward or punishment-relevant stimuli [47]). Previous work has identified that satiety improves set-shifting, while decreasing sensitivity to reward in humans [48,49,50]. However, past research has relied on different behavioural outcomes (i.e., reaction time) or self-report for both set-shifting and reward-sensitivity, both of which do not consistently correlate with cognitive measures of disinhibition [51]. Since food-relevant rewards are often more salient than monetary rewards under fasted conditions [52], the monetary rewards in the CIS may have been insufficient to elicit changes in this mechanism [53].

Of the appetite-related hormones, only LEAP-2 was significantly correlated with performance on the CIS, specifically, lower LEAP-2 was related to slower reaction times. Although research into the molecular pathways of LEAP-2 is still in its infancy, a rodent model has recently shown that LEAP-2, akin to ghrelin, acts on peripheral and central growth hormone secretagogue receptor (GHSR) [54]. Current data shows that LEAP2 acts as an antagonist at the GHSR [17], suggesting that it acts to dampen molecular pathways activated by ghrelin [55]. However, the significant relationship between high LEAP-2 and increased impulsivity in our human study is at odds with rodent data, wherein ghrelin infusions foster impulsive responses [56]. Provided that high plasma LEAP-2 should block ghrelin action at the ghrelin receptor, the observed relationship between LEAP-2 and impulsivity was unexpected. Nevertheless, this is consistent with the noted cross-species differences in ghrelin activity seen in animal models and humans [57,58]. Whereas animal models may rely more closely on hormonal signals as clear-cut triggers of fast responses to counteract homeostatic imbalance, in humans these signals could convey a need for accuracy rather than speed, or a low-level signal that competes with higher-order drives, prompting less reflective responses [59]. Furthermore, differences in methodology, such as a predominance of ecological field work in animals versus experimental studies in humans, or even the differences in experimental paradigms in human versus non-human animals may also contribute to discrepant findings [2,3,4,60,61,62].

A strength of this study is that it is the first to examine the cognitive drivers of hunger-related impulsivity in humans using a novel comprehensive measure. Unlike previous research that often looks at the multiple drivers of impulsivity in isolation (i.e., [11]), the use of the CIS allowed for the concurrent measurement of three empirically valid components of disinhibition. Moreover, our identification of a condition x self-reported hunger interaction for information-gathering performance indicates that subjective hunger may be an important predictor of impulsivity following meal consumption. Therefore, information-gathering/reflection may provide a useful cognitive treatment target in populations where subjective hunger may remain elevated following high-calorie consumption (i.e., obesity, binge-eating disorder [63,64,65,66]).

A limitation of the study is that we were unable to detect acyl-ghrelin from the plasma samples, however, we were able to successfully extract LEAP2 as a biological measure of hunger. LEAP-2 was originally identified from human blood ultrafiltrate [67], and was serendipitously discovered as an inverse agonist at the ghrelin receptor (GHSR). Although its biological function still remains poorly defined, recent studies in mice and humans demonstrate that LEAP-2 is acutely secreted into the circulation in response to oral glucose administration or a high calorie liquid meal (>600 kcal), and plasma LEAP2 is significantly correlated with post-meal changes in plasma glucose [17]. These results suggest that LEAP-2 is an acutely regulated satiety signal, which reduces ghrelin receptor activity after feeding. Our results are consistent with the notion that LEAP-2 represents a novel satiety factor, given that both higher LEAP-2 and satiety are independently associated with greater impulsive responses. Furthermore, this is the first study to link plasma LEAP-2 to behavioural traits, which represents a new avenue to study the integration of metabolic signals with complex behavioural traits in humans.

## Figures and Tables

**Figure 1 nutrients-13-02001-f001:**
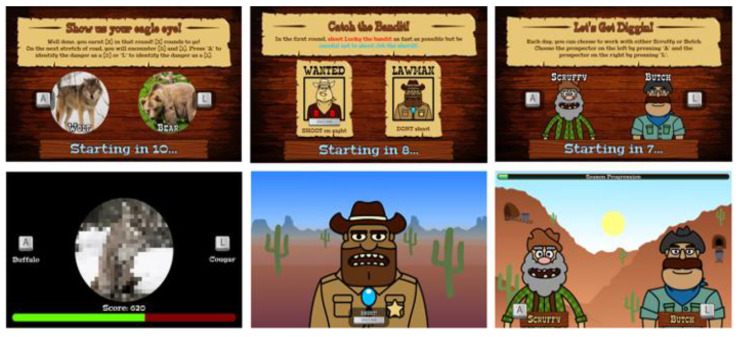
The three tasks from the CIS (from left to right): Caravan Spotter, Bounty Hunter, Prospectors Gamble. Presentation order of task components was counterbalanced across participants. Example trials for each task are represented as short clips on YouTube: https://www.youtube.com/playlist?list=PLs-GMH-Foyaz-UkhauX-uSCDBQkAM1sTv. Access date: 31 May 2018. Figure was adapted from a previous publication from our group [16].

**Figure 2 nutrients-13-02001-f002:**
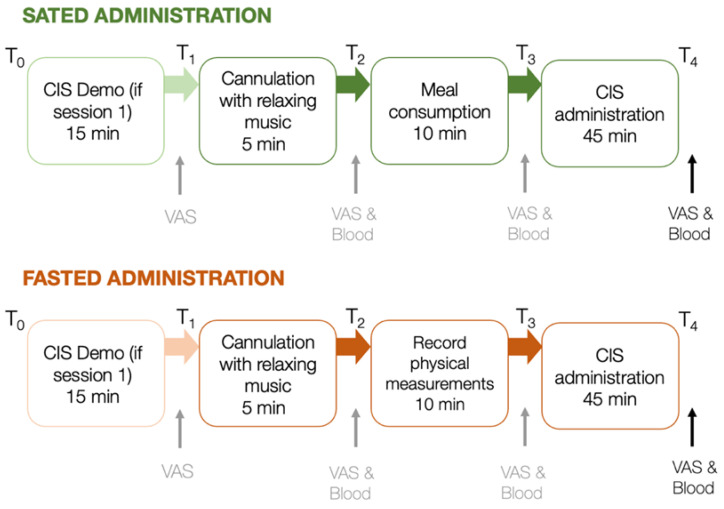
Procedure of laboratory protocol for sated and fasted sessions, scheduled 7–10 days apart. VAS and blood samples are taken at points indicated. VAS and blood data assessed at the last time point of each session were considered for further analyses. min = minutes; ↑ indicates the administration of VAS and/or blood samples between key steps in the testing protocol.

**Figure 3 nutrients-13-02001-f003:**
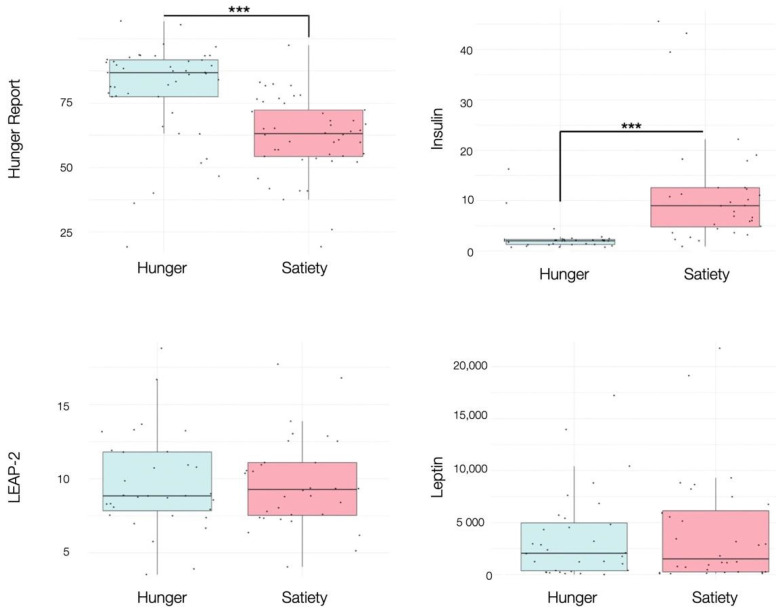
Boxplots showing the effects of hunger and satiety on subjective hunger reports, insulin (mU/L), LEAP-2 (ng/mL), leptin (pg/mL). See main text and Appendix A for statistics. *** *p* < 0.001.

**Figure 4 nutrients-13-02001-f004:**
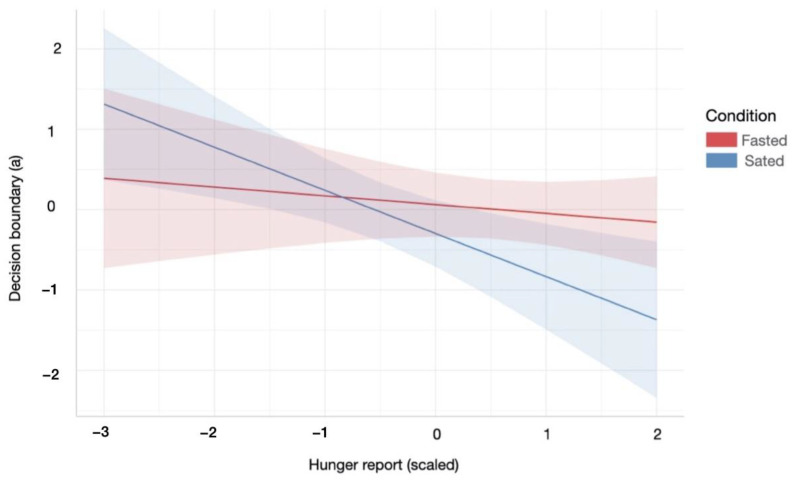
Relationship between the decision boundary parameter of the DDM and subjective hunger reports across hunger and satiety. Participants were less conservative in their information accumulation in order to make a decision during the satiety condition, with increased hunger levels. Red/blue coloured area indicates 95% confidence interval of linear model’s predictions. For statistical details refer to main text (Section 3.2.1.).

**Figure 5 nutrients-13-02001-f005:**
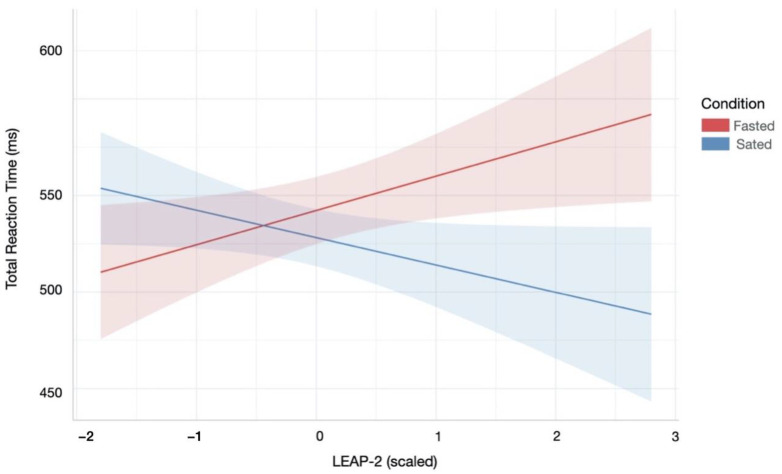
Relationship between reaction time during attentional control and LEAP-2 across hunger and satiety. Lower LEAP-2 levels during the sated condition were associated with slower reaction times overall. There were similar patterns for reaction times during correct and incorrect responses (not shown). Red/blue coloured area indicates 95% confidence interval of linear model’s predictions.

## Data Availability

Data presented in this study is available upon request from the corresponding author.

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
