# Peer review of "The Hunger Games: Homeostatic State-Dependent Fluctuations in Disinhibition Measured with a Novel Gamified Test Battery"

_nutrients, 2021, doi:10.3390/nu13062001_

Round 1
Reviewer 1 Report
The authors submitted their article titled “The hungergames: homeostatic state-dependent fluctuations in disinhibition measured with a novel gamified test battery”. They measured cognitive impulsivity, information gathering and attentional control along with blood measurements for insulin, leptin and LEAP-2, an antimicrobial peptide, in subjects that received, or not, a meal (satiety). They demonstrate that hunger levels during a satiety session are related to decreased information gathering. Specifically, hunger impaired decision-making and information gathering. These results were in line with the LEAP-2 blood measurements. The article is well-written and easy to follow. Some questions and suggestions are addressed below:
- Consider of adding “food” before “homeostatic states” in line 14 of abstract.
- Introduction: It is not clear what is the hypothesis regarding the LEAP-2 action.
- Did the authors consider smoking as a parameter of the demographics that could affect satiety?
- There is a section 2.3.1, but not a 2.3.2. This means it could be a separate section as it is for the 2.4.
- Could the authors provide for information regarding the meal used in their study?
- Figure 4: Please refer in which section statistical details are described.
- Section 3.2.3.: Is there a figure regarding these results?
- Discussion: Please provide possible molecular pathways that could support the results of LEAP-2.
- Does the LEAP-2 have non-specific actions on other receptors that coud potentially affect the authors results?
Reviewer 2 Report
Voigt et al. aimed to understand the impact of nutritional status (fasted vs. fed) on the disinhibition behavior/impulsivity in humans using a novel electronic game-based tests. The authors found that blood marker for fed state (LEAP-2) is significantly associated with faster reaction times in fasted condition, suggesting that residual hunger after food consumption may increase impulsive behavior. The manuscript was well written, and the study was executed properly with fixed meals. The findings extend our understanding of impulsivity aspects during normal hunger/satiety states and within the eating disorder pathophysiology.
Comments:
- What are blood/plasma glucose levels before and after the test sessions?
- There is no mention on Fig. 4 (at least the description and link to Fig. 4 was not clearly indicated). What is R2 and p value for the regression analysis?
